# Online Monitoring Technology of Metal Powder Bed Fusion Processes: A Review

**DOI:** 10.3390/ma15217598

**Published:** 2022-10-28

**Authors:** Zhuo-Jun Hou, Qing Wang, Chen-Guang Zhao, Jun Zheng, Ju-Mei Tian, Xiao-Hong Ge, Yuan-Gang Liu

**Affiliations:** 1School of Materials Science and Engineering, Xiamen University of Technology, Xiamen 361021, China; 23D METALWERKS Co., Ltd., Xiamen 361021, China; 3Engineering Research Center of Fujian University for Stomatological Biomaterials, Xiamen Medical College, Xiamen 361023, China; 4College of Chemical Engineering, Huaqiao University, Xiamen 361021, China

**Keywords:** additive manufacturing, powder bed fusion, online monitoring, selective laser melting, electron beam melting

## Abstract

Metal powder bed fusion (PBF) is an advanced metal additive manufacturing (AM) technology. Compared with traditional manufacturing techniques, PBF has a higher degree of design freedom. Currently, although PBF has received extensive attention in fields with high–quality standards such as aerospace and automotive, there are some disadvantages, namely poor process quality and insufficient stability, which make it difficult to apply the technology to the manufacture of critical components. In order to surmount these limitations, it is necessary to monitor the process. Online monitoring technology can detect defects in time and provide certain feedback control, so it can greatly enhance the stability of the process, thereby ensuring its quality of the process. This paper presents the current status of online monitoring technology of the metal PBF process from the aspects of powder recoating monitoring, powder bed inspection, building process monitoring, and melt layer detection. Some of the current limitations and future trends are then highlighted. The combination of these four–part monitoring methods can make the quality of PBF parts highly assured. We unanimously believe that this article can be helpful for future research on PBF process monitoring.

## 1. Introduction

AM (Figure 1) is a layered manufacturing technique. Compared with subtractive manufacturing technology, it can produce complex geometric shapes, lower the use of raw materials, and greatly reduce the cost. Currently, AM is widely used in aerospace, automobile manufacturing, biomedical, and other fields [1,2,3]. According to different feeding methods, metal AM is divided into two types: direct energy deposition and powder bed fusion [4]. The former adopts the method of synchronous powder feeding or wire feeding, and fills melt pool area with the raw material while scanning the high–energy beam; the latter adopts the method of laying the powder bed in the forming area in advance.

Metal PBF processes (Figure 2) comprise selective laser sintering (SLS), selective laser melting (SLM), and electron beam melting (EBM) [5]. In SLS, the process uses a laser beam as a heat source with high forming accuracy and surface finish. It has a wide range of materials that can form almost any geometrical part, especially for parts with complex internal structures. In EBM, the process uses a high–energy electron beam as a heat source, has faster forming speed and low forming thermal residual stresses, which can form high melting point material and brittle material.

Melting and depositing a powder bed by laser or electron beam is a highly dynamic and complex process with multiple physical phenomena and transformations. Metallic AM parts are prone to macro defects such as warping deformation [6,7], spheroidization [8], cracking [9], and internal defects such as porosity [10,11,12,13,14], incomplete fusion [15], and inclusions [16]. The timely detection and suppression of the defects in the formed parts can greatly improve the forming quality of the metal powder bed and eliminate the limitation of the technical instability of the process development.

**Figure 2 materials-15-07598-f002:**
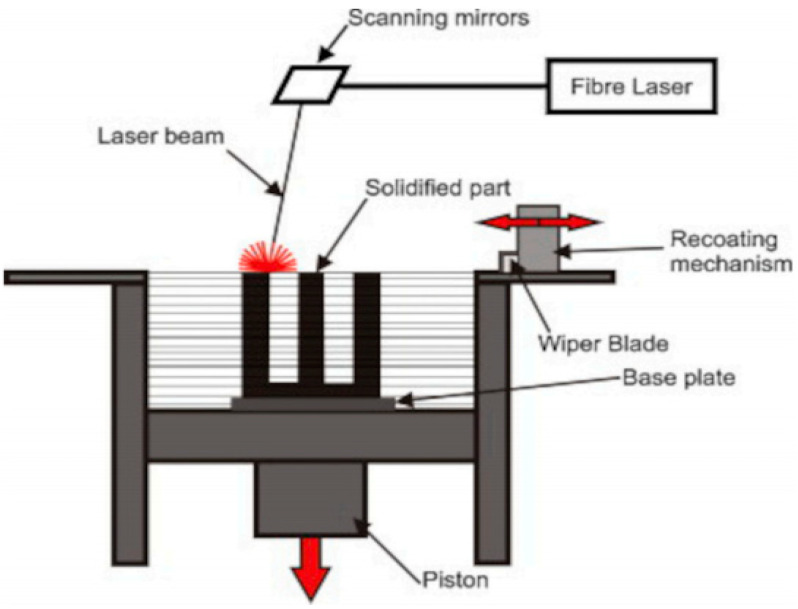
Schematic of the PBF process [17].

Online monitoring is a method of detecting defects in a timely manner to ensure the quality of parts manufactured by the PBF process. On the one hand, it can provide researchers with records and optimize process parameters; On the other hand, it can conduct real−time monitoring and data analysis on the process for online diagnosis, real−time defect repair, and provide key data for process records. Currently, AM is widely used in several fields. Compared with conventional manufacturing processes, parts formed by PBF are smaller in size and therefore more sensitive to defects. In recent years, many research teams have obtained a variety of online monitoring methods for PBF process. These monitoring methods have shown irreplaceable technical potential, but they are still in the research and development stage.

This article builds on previous studies and reviews methods for monitoring defects during PBF. Subsequentially, the monitoring results are analyzed, and the reliability, limitations, and improvement trends of the monitoring method are described. The paper is divided into six sections, including the present one. Section 2 surveys the monitoring of powder recoating, which is the first step in the PBF process and one of the key steps in stable forming. Section 3 discusses the powder bed inspection. Section 4 reveals the monitoring of building process. With a particular emphasis on monitoring the melt pool and temperature. Section 5 presents melt layer detection, namely the monitoring of temperature and surface topography. Section 6 summarizes existing work and identifies future research trends.

## 2. Powder Recoating Monitoring

Powder coating is the process of spreading powder on the forming area by a recoating device to form a powder bed [18]. It is the first step of PBF and one of the key steps for stable forming. For the faults that are prone to the coating process and abnormal damage to the coating machine, Berumen et al. [19] used a digital camera to monitor powder feed and coater defects during powder coating. Although this method is low cost, there are two major disadvantages. Firstly, it is necessary to correct the images taken, which often decreases the accuracy of PBF defect monitoring. Secondly, there is a trade−off between spatial resolution and the field of view. Reinarz and Witt [20] installed a piezoelectric accelerometer on coater of the SLM equipment (Figure 3) and monitored the powder coating process by measuring speed change signal of coater. When coater collides with previously deposited layer, it will cause large vibration or even jam. The signal of the accelerometer can reflect the smoothness of coating process and unevenness of previously deposited layer. Kleszczynski et al. [21] validated the reliability of the monitoring system by detecting defects with a high–resolution CCD camera. Experiments have shown that, with the help of image processing, the system can detect the defects such as powder deficiency, poor support, and damage to the coater. Process stability is monitored by studying critical process parameters and critical geometries. In the future, this approach can be used to build a knowledge base for specific materials, which can clearly understand the causes of defects and propose appropriate solutions. 

Liu et al. [22] proposed an on–site quantitative detection technology suitable for EBM and SLM. By evaluating the entire powder bed rear rake using edge projection profilometry, defects such as part thermal expansion, powder overfeed, and powder shortage can be inspected. The technique relies on a surface–fitting algorithm to calibrate the phase error during tracking to ensure the reliability of the tracking method. The results of the experiments show that the method can effectively inspect powder defects and use the results as feedback during the building process. The method has the advantages of low cost, fast acquisition time, and a no–vacuum environment. In the future, it can be considered for improving the processing speed and intelligent measurement of the calibration algorithm and realizing the automatic classification and feedback of defects. Seita et al. [23] revealed that powder layer defects in PBF systems (SLS, SLM, and EBM) and binder jetting can be detected with a high spatial resolution of ~5 µm using a powder scanner (Figure 4). By installing a line scanner on the coating machine, the two move synchronously to acquire the image of the powder bed, so this inspection technology has a lower cost and higher time utilization. Using an improved Laplacian algorithm to quantify out–of–focus areas in the image, the automated detection of powder bed defects (such as powder unevenness, ultra–high edges, and grooves) can be achieved. It is important to note that a slight difference between the movement speed of the coating machine and the sampling rate of the contact image sensor (CIS) in the powder scanner can cause image distortion.

## 3. Powder Bed Inspection

A powder bed formed after powder coating is the basis for scanning melted powder with an electron or laser beam. If the powder bed is not flat, it will cause the melt pool to become unstable during the scanning process and cause abnormal defects such as protrusions or voids that may affect subsequent forming. 

There are many reasons for the unevenness of a powder bed: the wear on and damage to the recoater, causing the powder bed to produce gullies or ridges distributed along the direction of the paving; recoater streaking occurs when the recoater blade drags a piece of debris or a tuft of powder onto the powder bed; the melt layer is raised, and the scraper is forced to jump at the bulge, resulting in vertical direction gully or bulge perpendicular to the direction of the paving; and an insufficient amount of powder or no powder at the end of the powder bed. Therefore, the appearance of the powder bed can not only reflect the working state of the powder coating device but also reflect the quality of the upper melt layer. The following are the methods available for monitoring powder bed defects in existing research. 

Figure 5 is a schematic diagram of the system for the visible light detection of a powder bed in the SLM process, including an optical camera and several flash sources. The system uses an off–axis arrangement where the camera aligns the powder bed laterally, while the flash source is tilted at different angles. The purpose of the flash source is to provide the appropriate background light to capture a clear, high−contrast image of the powder bed, simplifying the subsequent defect identification process. Craeghs et al. [24] used a visible light inspection system to detect powder bed defects due to damage or wear of the coater. Some grayscale distributions perpendicular to the powder coating direction were extracted from the grayscale powder bed images, and the average distribution was compared with a reasonable gray level to effectively identify errors and material discontinuities at the powder bed level (Figure 6). The coating is linear in this method, so it is not possible to accurately identify a defect at a certain point. Kleszczynski et al. [21] and Jacobsmühlen et al. [25] performed threshold processing on grayscale images based on the characteristics of bright areas generated by bulges in the melt layer to achieve the effective extraction of protrusions. In addition to powder bed defect detection, the method can also be used for parameter optimization (material identification). Jacobsmühlen et al. [26] provided an image−based two−dimensional acceleration method for investigating the influence of the angle of the suspended structure and the parameters of the supporting structure on the bulge melt layer. The method can rank the stability of different components and determine the best parameters for guaranteeing their stability. In the future, the approach could be used to guide the building design and validate the boundaries of the established parameters. Abdelrahman et al. [27] extracted the area corresponding to the section of the part from the powder bed image and superimposed it to form a three−dimensional powder bed model, which can well monitor the precise location of the powder bed anomaly corresponding to the part.

Neef et al. [28] proposed the use of low−coherence interferometry to detect the flatness of a powder bed in the SLM processes. The principle of low–coherence interference imaging is to scan the powder bed by measuring the laser beam, to measure the difference in an optical path between the reflected light and the reference light through the spectrometer, and to compensate for the deviation caused by the angle deflection to obtain the height distribution of the different scanning points. It can be seen from Figure 7 that low–coherence interference technology can effectively detect the height and low fluctuation of the powder bed and can identify the groove of 50 µm depth on the powder bed. Fleming et al. [29] revealed inline coherence imaging (ICI) tracking topography, providing the instant inspection of surface roughness, damage to the recoater blade, and powder–packing density. The level of energy during 3D build affects surface roughness, which can be corrected based on ICI measurements. Moreover, the method successfully realizes manual closed–loop control and full feedback control. Boschetto et al. [30] proposed the use of digital image processing to monitor the defects of powder beds during SLM. In this study, thousands of images of powder beds taken by CCD cameras were analyzed using 2D and 3D analysis to identify single−layer defects and defects between powder beds, respectively. Using this method, the location and size of defects can be found accurately and quickly and at a low cost. However, the drawback of this method is that the image may be distorted and must be calibrated.

The above optical detection has many limitations in the EBM process, so they are mainly aimed at the SLM process. These limitations can be seen from the following: The detection method based on optical imaging has strict requirements for the location of sensors and light sources, so the molding equipment needs to be modified accordingly, thereby increasing the difficulty of system integration. The inert gas in the forming chamber of SLM can inhibit the evaporation of metal, providing good heat dissipation conditions, so that the sensor can be directly put into the forming chamber, which makes the system integration simple. However, the EBM process provides a vacuum environment, and the metal evaporation in EBM cannot be solved well, so the ambient temperature is very high, and the radiation is very strong.

Li et al. [31] revealed an enhanced phase−measurement profiling technique (EPMP) to monitor defects such as inhomogeneities in powder beds. This method can significantly improve inspection efficiency while having high accuracy compared with the conventional phase−measurement profiling technique. In addition, this method can be used to monitor fusion area defects. Currently, the technique does not allow for real–time closed–loop control and automatic defect identification and classification.

Grasso [32] investigated a method suitable for monitoring powder defects in the EBM process. This method acquires images by layering them with a camera. Image processing is then used to identify inhomogeneities in the powder layers. It is experimentally demonstrated that the technique can currently detect three defect anomalies, namely, powder deficiency, powder overload, and powder bed contamination. Compared with the method of streak projection field measurement in the abovementioned article, this method has a higher data collection rate because of the layered image acquisition. This method also has a high practicality and can be readily used in industrial−grade EBM devices. Currently, the method cannot be used to monitor surface powder oxidation.

The monitoring methods used above usually only target specific defects in the powder bed and require the use of high–resolution cameras. Therefore, the entire powder bed is usually not monitored, which has limitations. In order to improve detection efficiency, machine learning (ML) algorithms have attracted people’s attention. To verify that ML can properly and accurately track defects in AM processes, numerical simulation is necessary because of the high cost of AM technology. Jahan et al. [33] modeled the LPBF process by computer hydrodynamics and generated a large number of simulated values. The simulated values are entered as input data into a graphics−based neural network machine learning algorithm to monitor defects in LPBF processes. Experimental results show that this machine learning algorithm can predict the defects caused by thermal anomalies in PBF processes. In addition, feedback control is carried out by optimizing process parameters to reduce the formation of defects.

Xiao et al. [34] proposed a method for monitoring powder bed defects using the time–spatial convolutional neural network (TSCNN) model, which is suitable for different systems of PBF (SLS, SLM, and EBM). Start by taking an image of the powder bed with a digital camera. The image is then split into three monochromatic channels corresponding to the RBG. Finally, TSCNN and the region proposal network are utilized to detect powder bed defects in selective laser sintering. Compared with other methods for detecting defects, this approach has higher accuracy and efficiency while resisting geometric distortion and image blurring. In the future, surface defects can be detected for other related applications and can also be used to set machining parameters in real time.

Scrime et al. [35] presented a multiscale convolutional neural network (MSCNN) based on a machine learning algorithm for the automatic, high−accuracy classification of anomalies detected during powder bed monitoring (coater jumps, recoat streaks, and debris). Figure 8 shows the powder bed image taken on the 2709 floor. L. Scrime and J. Beuth used MSCNN technology to analyze the anomalies detected in the 2709 layer and found that the recoil jumper (blue−green) and partially damaged (magenta) and incompletely diffused (yellow) fragments were captured, as shown in Figure 9. In the future, anomaly classification accuracy can be further improved by incorporating additional data in the neural network. In addition, the real–time analysis of high–resolution images is performed by designing a neural network structure.

Scrime et al. [36] proposed to monitor and classify defects in powder layers using ML algorithms. The powder layer image captured by the digital single−lens reflex (DSLR) camera was fed into the ML model. An algorithm is used to classify images so that similar images are grouped. In this experiment, the authors detected six different powder layer anomalies: recoater jumps, recoater streaks, debris, superelevation, part failure, and incomplete spreading. Although this method reduces the amount of computation, it is less accurate for monitoring recoating streaks due to the small amount of training data. 

## 4. Building Process Monitoring

During melting, a high−energy electron beam or laser beam melts powder to form a melt pool that is solidified and deposited to form a solid cross section. Therefore, the fused deposition process directly determines the quality of the final melt layer. Currently, the monitoring of the melt process is primarily directed at the melting pool temperature and the entire forming area.

### 4.1. Melt Pool Monitoring

In the PBF processes, the melt pool is formed by laser beam or electron beam scanning of the powder bed and has the characteristics of small size and fast–moving speed. The quality of the castings is greatly influenced by the shape, size, and temperature of the melt pool. Melt pool monitoring is the real−time measurement of radiation intensity and shape characteristics during laser or electron beam scanning and the real–time analysis of measurement data to identify spheroidization and warpage.

Melt pool monitoring usually uses a coaxial layout, including that of Berumen et al. [37]. As shown in Figure 10, the sensing channel overlaps with the shaped laser beam path in order to acquire the melt pool signal in real time without adding a complicated melt pool tracking system. The high–power forming laser beam enters the scanning system after being reflected on the surface of the 45° half mirror, and the melt pool radiation signal propagates in the opposite direction. After passing through the half mirror, the filter is used to filter out the specific band signal into the sensor or through splitting. The mirror splits light into two beams for sensor acquisition. Compared with traditional cameras, the method has high timeliness and high local spatial resolution. Moreover, the method enables the real–time control and recording of the manufacturing process. The research groups Kruth et al. [38], Clijsters et al. [39], and Craeghs et al. [40] used coaxial sensing to screen the 740~950 nm radiation wave and split it. One beam is used for the photodiode to collect the pool intensity signal; the other beam is imaged by a high–speed CMOS camera to extract geometric information such as melt pool area, length, and width. The light intensity of the pool collected by the photodiode is proportional to the size of the pool. In addition, the size of the melt pool can be kept constant by controlling the laser power. Both of them can effectively detect process problems such as spheroidization during the forming of suspended structures, convex hulls at corners during U–shaped scanning, and powder coating failures. The research groups used laser power as the control object and used the output voltage of the photodiode and the pixel area of the melt pool as feedback variables to establish a closed–loop control system for SLM. The research results show that the above two closed–loop control systems can effectively improve the accuracy of forming surface structures formation and realize the adaptation of process parameters. Kanko et al. [41] applied a coaxial optical path to low–coherence interferometry to measure the height of a melt pool in real time during the SLM forming process. Figure 11 shows that when the laser beam is swept over the suspended area, the melt pool overheats due to poor heat dissipation conditions, which causes severe fluctuations. Low–coherence interferometry can quickly capture changes in melting pool height and identify globular defects caused by overheating.

In order to facilitate defect identification and positioning, Clijsters et al. [39] implemented location–based visual pore detection. The monitoring system uses photodiodes to monitor the strength of the molten pool and uses a near–infrared thermal CMOS camera to monitor the area of the molten pool. Data monitored by both are visualized through a mapping algorithm. The pixel area of the melting pool is arranged into two dimensions according to the scanning position, and the time series signal is transformed into a spatial distribution image. The geometric parameters of the steady−state melt pool during a filling scan and contour scan were obtained experimentally and used as reference data. In addition, by obtaining the position−dependent melt pool area distribution layer by layer, the three−dimensional spatial positioning of the internal pore defects of the formed part is realized. The experimental results show that the defect identification and localization results are in good agreement with the computed tomography results. This method has a high sampling rate and can be monitored in real time, but no feedback control is currently implemented. In the future, automated detection can be achieved by adding algorithms. Krauss et al. [42] proposed the use of thermal imaging to monitor defects (pores) and irregularities (irregularities close to overhang structures) in SLM processes due to insufficient heat dissipation using an off−axis sensor arrangement to align the infrared camera from the front observation window of the SLM device to the forming area and take a thermal image during the laser beam scanning. By processing the infrared thermal image, geometric parameters such as the area of the melt pool, aspect ratio, and roundness are extracted. The effects of process parameters such as scanning speed, laser power, hatch spacing, hatch length, and powder layer thickness on the geometric parameters of melt pool were also studied. Since the reference value has a certain relationship with the moving direction of the current process area, there is a parameter deviation in this method. This technique will be explored in the future for monitoring coater wear and powder layer thickness. Feedback control could not be achieved with the above methods. Le et al. [43] revealed a method for monitoring the melting pool scale in PBF processes using CMOS cameras. In this method, the experimental measurement data and the simulation results are compared, and the results show that the numerical simulation can predict the size of the melting pool with a small error. This provides great help to the feedback control of parameters in the process of melting pool monitoring to avoid defects to the greatest extent possible.

The abovementioned method for monitoring melt pools basically uses a coaxial system in SLM processes since coaxial monitoring can track melt pools well, and the output signal is simple. Real–time process monitoring and feedback control have been implemented. In the case of an on−axis setup, the laser can be affected by lens characteristics in the Lagrangian reference frame. In the EBM process, due to structural limitations such as the electromagnetic deflection system, only the off−axis arrangement can be used, and the rapid tracking of the melt pool becomes a problem. Coupled with severe evaporation effects, the real–time monitoring of the melt pool is difficult.

### 4.2. Temperature Monitoring

PBF belongs to a kind of thermal processing. Recording and analyzing its temperature change process is of great value for understanding the inherent mechanism of the process and verifying simulation models. Price et al. [44] used near–infrared thermal imaging equipment to study the temperature distribution in the forming area during preheat scanning, contour scanning, and filling scanning in the EBM process. The use of a thermal imaging system in the above method does not give an accurate actual temperature. In order to make the temperature more accurate, Pavlov et al. [45] believed that the temperature of the laser impact zone could detect the changes in the SLM parameters. The method adopts a coaxial arrangement scheme, using a two–color pyrometer to measure the temperature signal of the melt pool during the laser scanning in real time. It is found that the measured values of the two–wavelength pyrometer are very sensitive to the process parameters such as filling interval, the thickness of powder layer, and filling strategy, but they cannot evaluate the integrity of the part.

Cheng et al. [46] and Price et al. [47] investigated the effects of process parameters such as the speed of scanning, electron beam current, and beam spot diameter over the longitudinal melt pool distribution (along scanning direction) as well as the melt pool size. The measured temperature distribution and the size of the melt pool were used to verify simulation model. In addition, Price et al. [48] and Gong et al. [49] explored the influence of forming height on the longitudinal temperature distribution of a melt pool and of the suspended surface at different distances from the centerline of the melt pool. It was found that longitudinal temperature distribution of the melt pool during filling scanning is repeatable and very sensitive to heat dissipation conditions. Finally, it is verified that the defect identification of temperature spatial distribution is feasible.

Internal void defects will weaken the heat dissipation ability of the local area and change the distribution and evolution characteristics of the surrounding temperature. Accordingly, Krauss, Eschey, and Zaeh [42] proposed a method for detecting internal void defects using temperature detection. When the laser beam sweeps through a predetermined defect area, the temperature distribution curve along the longitudinal direction of the melt pool is extracted. In comparison to the defect–free temperature distribution curve, the temperature distribution curve at the back end of the melt pool is found to have significant differences in defect position and size. Therefore, the dynamic data on the temperature of the powder bed can not only identify the pore defects but also obtain information such as the size of defects. 

Moreover, Krauss et al. [50,51] attempted to identify defects based on the time–domain evolution of the temperature. Relevant key indicators were extracted from the dynamic temperature evolution data, including high temperature holding time, equivalent thermal diffusion coefficient, maximum temperature, and splash amount. The equivalent thermal diffusion coefficient is the cooling rate defined by a one–dimensional downward thermal diffusion simplified model. The index extracted by each layer of temperature evolution can form a frame index map. After the forming is completed, distribution maps of the layers are stacked to form a three–dimensional index distribution model. This method has a layered nature. In the subsequent process, all the information needed for part quality can be obtained by monitoring each layer. Then, the quality index diagram of each layer is derived. Finally, a 3D quality report similar to the tomography method is combined. 

In terms of microstructure prediction, Price et al. [47] extracted the average cooling rate of shaped cross sections at different scanning speeds based on the temperature–time evolution curve and found that fast scanning produces higher cooling rates and smaller β columnar crystals. Raplee et al. [52] used different scanning strategies (point scan and line scan, as shown in Figure 12) during the EBM process. By analyzing these data, it is found that the thermal gradient and the velocity of the solid–liquid interface are approximately the same, so they are related to some changes in microstructure. The thermographic data were analyzed to determine the transition of material from metallic powder to a solid as–printed part. Compared with the experiment, it is found that the line scanning strategy will form a higher temperature gradient and a lower solid–liquid interface velocity, tending to form columnar crystals. The point scanning strategy will form a lower temperature gradient and a higher solid–liquid interface velocity, which is beneficial to equiaxed crystal forming (Figure 13). The above research shows that using thermal imaging to predict the microstructure of forming parts is helpful for achieving flexible preparation and effective control of forming parts in the process of adding materials.

Williams et al. [53] used a wide–field in situ infrared imaging system to monitor the powder surface temperature of the entire powder bed. This system studies the influence of interlayer cooling time by constructing cylindrical scenes with different heights. In the whole construction process, the in situ surface temperature data were obtained and compared with the results of porosity, microstructure, and mechanical properties. The research shows that using thermal imaging technology to predict the microstructure of the formed parts is helpful for controlling the part structure in the manufacturing process.

Although the real–time acquisition and analysis of dynamic temperature in the PBF process have made great progress in internal defect detection and tissue prediction, there are still many deficiencies. Due to the insufficient time resolution and spatial resolution of these thermal imaging devices, the accuracy and sensitivity of current defect detection are insufficient. Moreover, the EBM process produces a large amount of metal vapor, which makes it difficult to conduct continuous dynamic monitoring of the temperature. Although scholars have studied anti–vapor deposition systems and transmittance compensation methods, they still cannot completely eliminate the effects of evaporation [54,55]. In addition, in order to convert the radiation intensity output by the camera into an absolute temperature, it is necessary to accurately determine the parameters such as the infrared emissivity, window transmittance, and ambient temperature of the material, which also brings difficulties and challenges to real–time temperature measurement [56,57].

## 5. Melt Layer Detection

When a powder bed is melt–deposited by electron beam or laser beam to form a melt layer, the state of the melt layer not only reflects the quality of melt deposition and the matching of process parameters but also affects the subsequent coating and melting forming processes. Therefore, melt layer detection is a very important part of online monitoring. It can detect cross–section profiles, geometric parameters, and surface defects. At the same time, it can record the forming results of each layer and provide basic data for final quality traceability. At present, the main objects of melt layer detection are temperature and surface topography morphology.

### 5.1. Temperature Detection

The temperature detection of the melt layer is similar to the temperature detection in the fused deposition process. Both of them use a near–infrared/infrared thermal imager. The difference is that the temperature of the melt layer changes slowly. Usually, temperature is taken only once to extract and identify part contour defects in a single frame image. Schwerdtfeger et al. [58] used different focus bias parameters in the EBM process as a control. The thermal image of the melt layer was compared with the metallographic diagram. The results show that thermal imaging can effectively reveal unmelted material and defects within the cambium. Dinwiddie, Dehoff, Lloyd, Lowe, and Ulrich [54] used infrared thermal imaging to study the influence of EBM focusing parameters on the porosity and evolution of subsequent melt layers on suspended surface. Rodriguez et al. [56] used temperature distribution histograms to quantify the nonuniformity of the temperature distribution of the melt layer and found that temperature distribution of the superheated region is wider. It was considered that histogram analysis was a method for effectively identifying local overheating defects. Ridwan et al. [59] realized the extraction of workpiece sections through image processing and calculated the porosity of the melt layer to reflect quality. Mireles et al. [60] verified the feasibility of online defect repair by infrared thermal imaging monitoring. Compared with the thermal images before and after the remelting of melt layer, remelting effectively reduced pore defects. Based on this, the authors also proposed a closed–loop control method for the online repair of local defects.

The above research demonstrates the feasibility of using near–infrared or infrared thermal imaging technology to detect melt layer defects. Real–time melt layer detection based on this technology needs further research.

### 5.2. Surface Topography Detection

#### 5.2.1. Optical Inspection

The aforementioned powder bed detection techniques, such as visible light imaging and low coherence interference, can also be applied to detect the morphology of a melt layer. Foster et al. [6] extracted contours from melt layer optical images and stacked them to form a three–dimensional solid model. The three–dimensional model not only contains size information of the formed part but also visually shows the problem of the uneven powder bed (Figure 14). Abdelrahman et al. [27] took five images of a melt layer under different lighting conditions after each layer scan. First, the melt layer contours are extracted according to a CAD model of the part. The front and back three–layer contours are averaged, and these images are segmented to obtain cross sections. The cross sections are stacked layer by layer to obtain a three–dimensional solid model. Then, an abnormality occurring in the same position on at least two adjacent melt layers is regarded as a real defect. Finally, the internal parts such as unfused material and voids were identified and located. DePond et al. [13] studied the influence of melt layer surface roughness under different filling strategies. Low–coherence interferometry was used to monitor change roughness when forming a suspended structure. Compared with the height distribution map, it was found that the melt layer has a greater roughness and a clear directionality under round–trip scanning without rotation between layers. When the uneven layers are accumulated layer by layer, the entire part will eventually be deformed. In addition, Erler et al. [61] proposed a monitoring method for measuring height distribution surfaces using 3D mapping technology and studied the influence of coating parameters and laser power on uneven melt layers. With this method, the uniformity, thickness, and layer defects of powder layers and sintered layers can be monitored. To avoid false detections during the detection process, direct process control should be added to the detection method. Imani et al. [62] used X–ray computed tomography to identify pores and obtained layered optical images of powder layers during part manufacturing. Then, spectral theory and multifractal features were extracted from the layered images of each test part. Finally, the machine learning method was used to link these features with the process parameters (Figure 15).

By measuring the laser beam to scan melt layer line by line, the laser displacement sensor receives reflected signals and calculates height at different positions. The result is a standard for calculating the height distribution of a melt layer, and the quality is judged according to this standard. Monitoring the melt layer using visible light imaging technology is difficult due to the analysis and processing of the grayscale images. Currently, contour extraction and defect recognition have been implemented, but most of this technology uses offline processing; the output of low–coherence interference imaging and 3D topography mapping is the distribution of melt layer heights, which reduces data analysis and processing, but its problem is that it must be scanned point by point or progressively, which increases the time cost. Moreover, due to the complex detection system, there are not many detection studies that directly measure height distribution.

#### 5.2.2. Electro–Optical Inspection

Electro–optical inspection is a unique technology for monitoring the morphology of the melt layer in EBM. Because electron imaging and scanning electron microscopy (SEM) have the same advantages, they can also be used in the current EBM detection technology. As electronic imaging is an image generation mechanism, it is not affected by the above problems such as thermal cameras and optical cameras. Watt [63] showed that after scanning layer by layer, a small electron beam was used to scan the molten layer point by point. The secondary electron and backscatter electron signals were collected and arranged into two–dimensional grayscale images according to scanning point order. Figure 16 is a schematic of the principle of electron microscopy. Electron detectors in existing studies are usually placed below the electron beam, above the building platform, and coaxial with the electron beam. Metal plates are often used as detectors due to the adverse environmental effects of high evaporation, high radiation, and high temperature. 

The use of electro–optical imaging to monitor the surface topography of a melt layer can greatly reduce difficulty in extracting the contours of the melt layer and identifying defects. Due to differences in the morphologies of the powder region and the solid region, the two can be easily separated to extract cross sections, measure contour size, and create 3D reconstructions based on the electron–optical image; Furthermore, the number of backscattered or secondary electrons emitted by the sub–micron pore area is small, and they appear as dark spots on the electron optical image, so it is easy to identify and locate pores. Arnold et al. [64] found that electro–optical monitoring can effectively identify pore defects and gives information about the quality of the resulting components. Wong et al. [65] separated the powder layer and the molten layer using electronic images and used specially designed hardware to detect the interaction between the electron beam from the machine and the treatment area, thereby generating a digital electronic image. This method is simple and reliable, so it can be a very good system for establishing rapid process optimization and timely feedback. Wong et al. [66] proposed an index to estimate the spatial resolution including the information depth of backscattered electrons (BSE), and they estimated the spatial resolution that can be achieved by Arcam A1 EBM electronic imaging. The experimental results show that the spatial resolution is 0.3 to 0.4 mm at room temperature. This study is helpful for the quality evaluation of on–site monitoring EBM process.

Electro–optical monitoring effectively overcomes the difficulties of high temperature, high evaporation, and strong radiation in the EBM process, greatly simplifying the difficulty of online monitoring and feedback control. Although electron optical monitoring has initially achieved defect recognition and feedback control, much research work is still needed for the analysis and interpretation of electro–optical images. In addition, in the existing research, electro–optical images are insensitive to macroscopic topography information, such as undulations and roughness, and it is impossible to extract effective macroscopic morphological information from them. Further research is needed in the detection of macromorphologies. 

#### 5.2.3. Acoustic Inspection

In the AM process (PBF), as with optics and thermals, acoustic sensing is also regarded as a key technology. Rieder et al. [67] used ultrasonic testing (UT) technology to monitor porosity. In the experiment, changes in laser power induce the formation of the porous layer. When ultrasonic waves enter the porous layer, the detector detects the reflection and scattering of the wave, thereby generating porosity. Compared with X–ray computed tomography (CT), UT can correlate online and offline data. Figure 17 shows the ultrasound scan and X–ray CT images. The technique has certain limitations and is currently limited to parts with simple geometries. In the future, the method will be used to classify defects.

Ye et al. [68] showed that there is some connection between the acoustic signal and the laser power. Using a deep belief network (DBN) to simplify the monitoring steps of the support vector machine and monitoring the PDF process by acoustic emission spectroscopy found that five different defect states could be detected, namely spheroidization, mild spheroidization, normal, mild overheating, and overheating. Smith et al. [69] presented spatially resolved acoustic spectroscopy (SRAS) to monitor porosity. However, because the signals generated in the PBF process are very complex, machine learning needs to be used to analyze these signals. Shevchik et al. [70] revealed that acoustic emission has a faster processing speed than imaging and tomography. The method combines acoustic emission spectroscopy and convolutional neural networks to pinpoint the locations of defects and monitor the porosity of parts. Furthermore, samples can be classified by porosity grade. In this study, the SLM process was monitored, and its feasibility was verified. In the future, this technology can be migrated to other PBF process monitoring.

## 6. Perspective

Laser selective melting and electron beam selective melting have significant differences in their monitoring. Optical monitoring technology has developed rapidly in the former for melt pool dynamic monitoring, powder bed inspection, and melt layer detection, and melt pool dynamic monitoring has been applied to process feedback control. In the field of electron beam selective melting process monitoring, optical monitoring technology is severely restricted, but electron optical imaging has become an effective means of reliably monitoring the forming quality of the electron beam selective melting process. 

The following is a summary of the various parts of this paper. Table 1 summarizes the powder recoating monitoring process. Table 2 provides an overview of powder bed inspection. Table 3 reveals building process monitoring. Table 4 presents melt layer detection. Analyzing Table 1, Table 2, Table 3 and Table 4, the monitoring methods summarized in this paper can detect some defects in the PBF process in time, but there are some common weaknesses:

(1) The monitoring method in this paper detects only superficial defects.

(2) Much of the data processing is offline, and the analysis of monitoring results relies on empirical data.

(3) All are measured by a single monitoring method, which may have an impact on monitoring stability.

(4) Conventional online monitoring techniques for powder–bed melting, such as those based on optical imaging, are limited by a number of factors.

(5) Some machine–based monitoring methods are less reliable.

(6) The monitoring method in this paper is only used to monitor defects in operation and does not involve the correction of defects detected.

Additionally, the authors believe that the online monitoring of the PBF process has the following development trends:

(1) Gradually from monitoring surface state to monitoring internal defects and grain morphology. The online monitoring technology of early powder bed melting intends to monitor the macroscopic morphology of parts by measuring the radiation intensity of the melting pool. With the development of online monitoring technology, the online monitoring of defects in parts has become a popular topic and involves the monitoring of grain structure, which will lay the foundation for the realization of real–time defect repair and organization control.

(2) More automated and intelligent. Most of the above online monitoring studies use offline data processing methods, and the analysis of monitoring results depends on empirical data. With the deepening of research, through the introduction of computer vision, artificial intelligence, data mining, and other technical means, defects and pores can be monitored more accurately and efficiently to further promote the development and application of online monitoring technology. 

(3) Multi–information fusion monitoring. Use multiple monitoring methods to monitor different stages of the process. Comprehensively judge the stability and defect information of the forming process according to different monitoring data. This monitoring technique not only makes up for the deficiencies of a single measurement method but also avoids the uncertainty of a single–signal indicator. By realizing the comprehensive processing and judgment of multi–sensing signals and multi–physical information of the process, the exactitude and reliability of the monitoring system are improved. 

(4) Active online monitoring. The conventional online monitoring of PBF processes, especially the monitoring technology based on optical imaging, is subject to many restrictions, namely lighting, metal evaporation, and temperature. Active monitoring technologies (such as low–coherence interference and electro–optical imaging) actively emit measurement beams or electron beams, which not only reduce the requirements on the working environment but also improve the sensitivity of the measurement system, the ability to counteract interference, and the adaptability to the working environment. Active monitoring is a more promising direction in the future online monitoring of PBF.

(5) Real–time and interactive data. A large amount of data will be generated in the abovementioned monitoring method based on machine algorithms. Due to the closed–loop control system, feedback from the online control process is usually not realized. Therefore, some monitoring methods based on machine algorithms have poor credibility. Improve data interaction by establishing a unified data communication protocol. The online monitoring technology based on machine algorithms obtains real–time feedback during the monitoring process, which improves the accuracy and efficiency of the monitoring system.

(6) Correction of observed deficiencies. These monitoring methods monitored deficiencies at different stages of the process, but they did not correct the deficiencies observed. According to the defect information provided by the monitoring here and some process parameters, there will be a good trend in the future research on defect correction methods.

## Figures and Tables

**Figure 1 materials-15-07598-f001:**
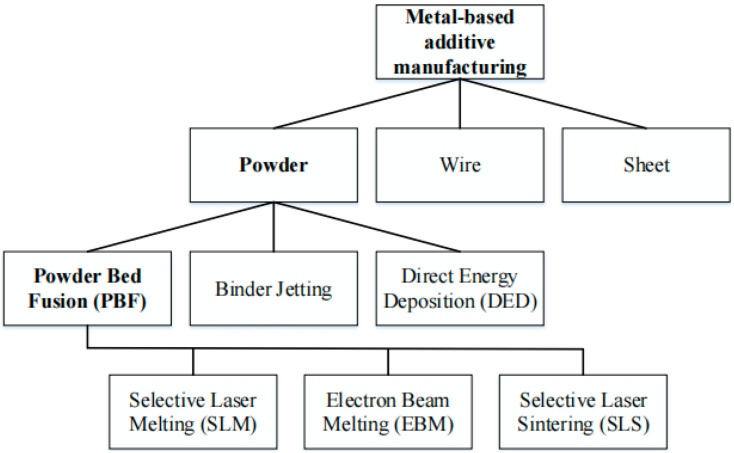
Overview metal−based additive manufacturing.

**Figure 3 materials-15-07598-f003:**
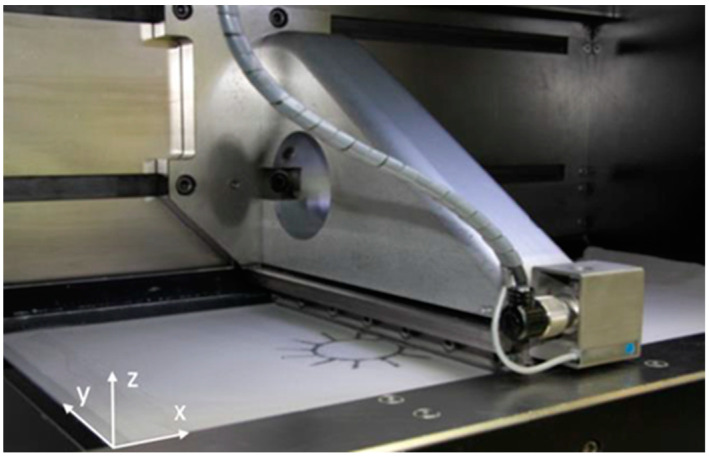
Integration of acceleration sensor at the recoating mechanism [20].

**Figure 4 materials-15-07598-f004:**
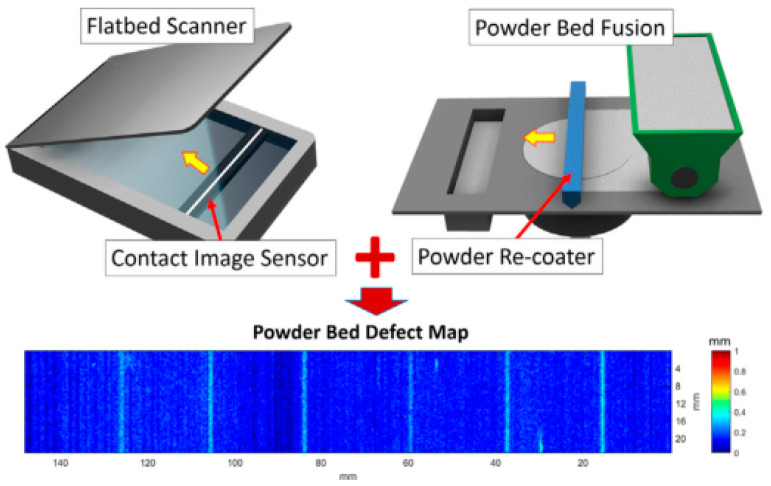
Components of powder scanner [23].

**Figure 5 materials-15-07598-f005:**
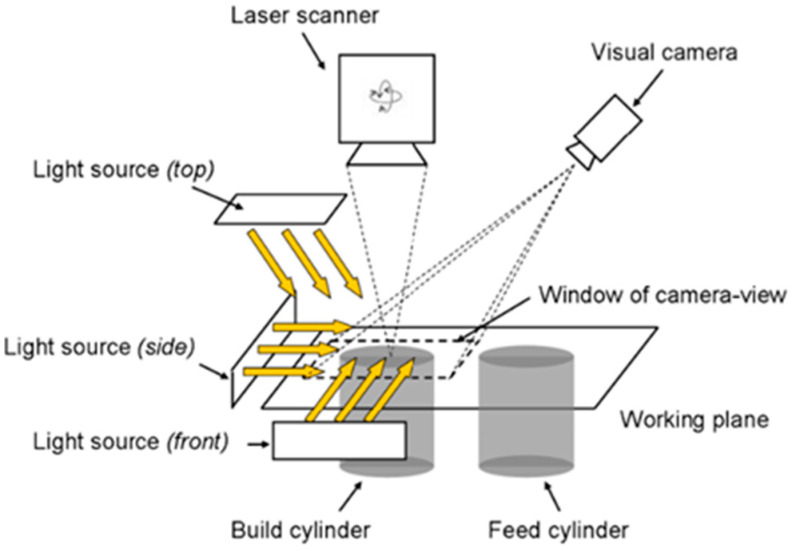
Visible light inspection system for powder bed [21].

**Figure 6 materials-15-07598-f006:**
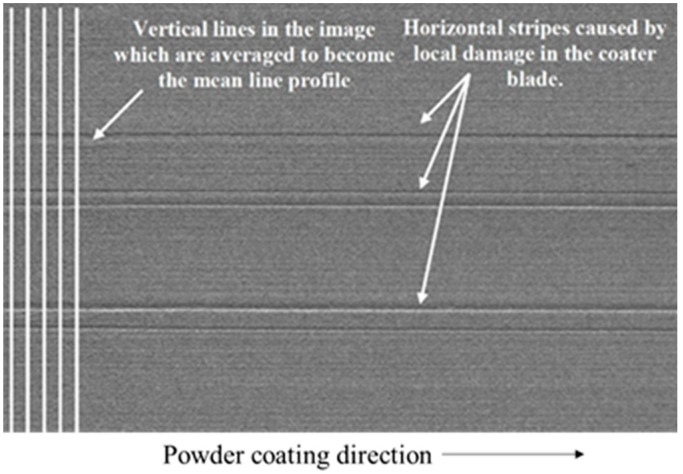
Image of deposited powder bed with worn coater blade [21].

**Figure 7 materials-15-07598-f007:**
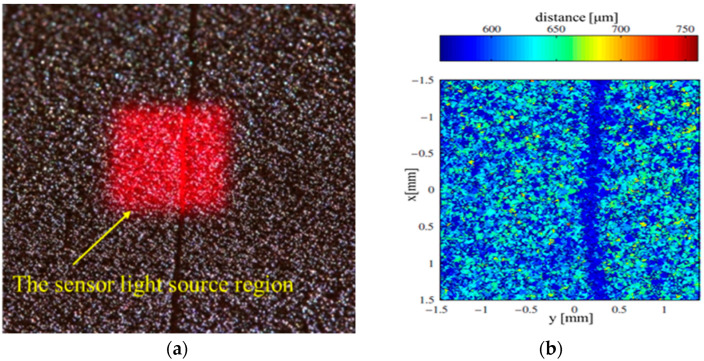
Powder bed inspection using low–coherence interference imaging in SLM. (**a**) Optical morphology of powder bed surfaces; (**b**) Profile scanning of powder materials [28].

**Figure 8 materials-15-07598-f008:**
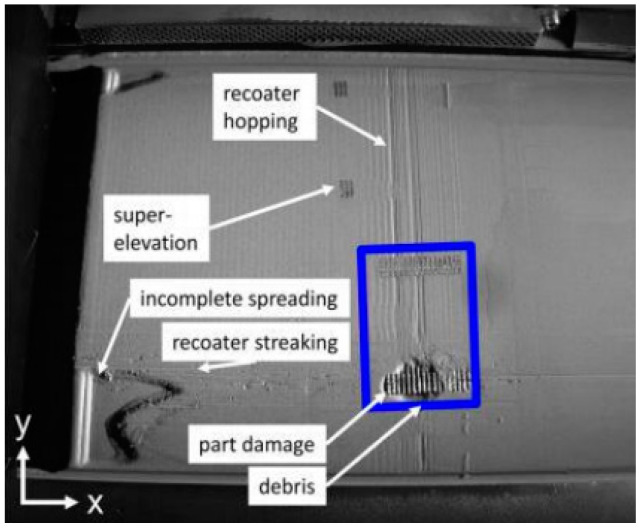
Anomaly diagram of powder bed [35].

**Figure 9 materials-15-07598-f009:**
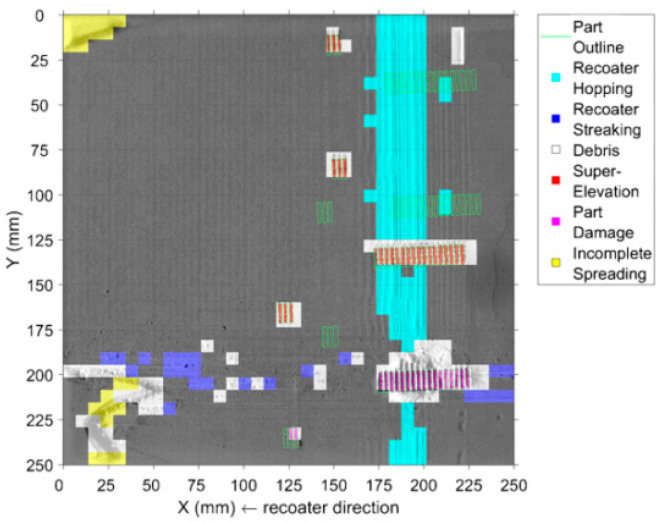
MSCNN anomaly analysis diagram [35].

**Figure 10 materials-15-07598-f010:**
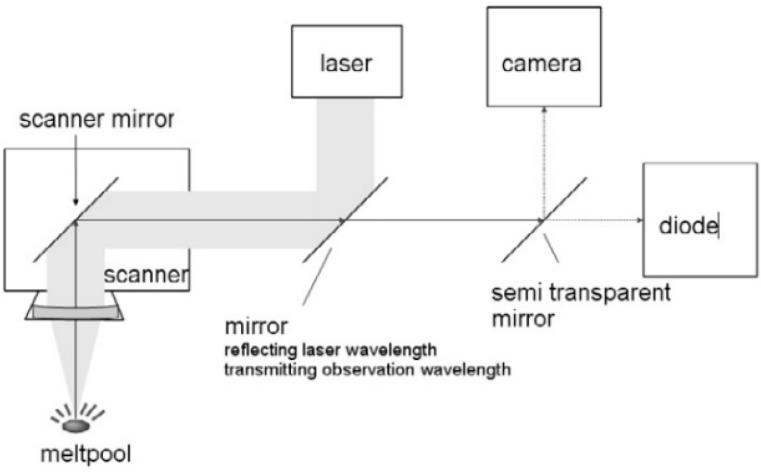
Schematic assembly of online process control [37].

**Figure 11 materials-15-07598-f011:**
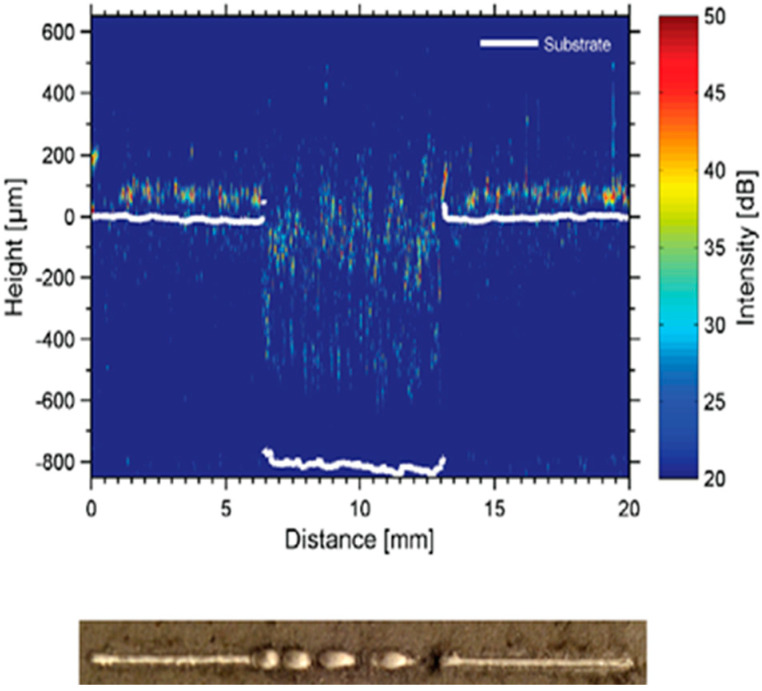
Single−channel SLM pool monitoring based on low coherence interference [41].

**Figure 12 materials-15-07598-f012:**
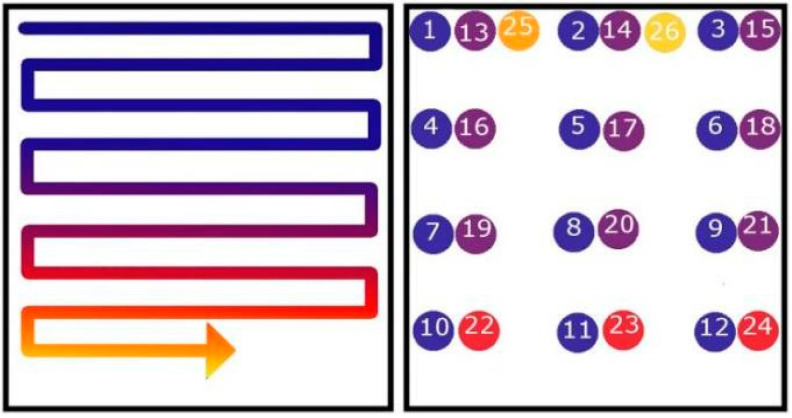
Line scan and point scan schematics [52].

**Figure 13 materials-15-07598-f013:**
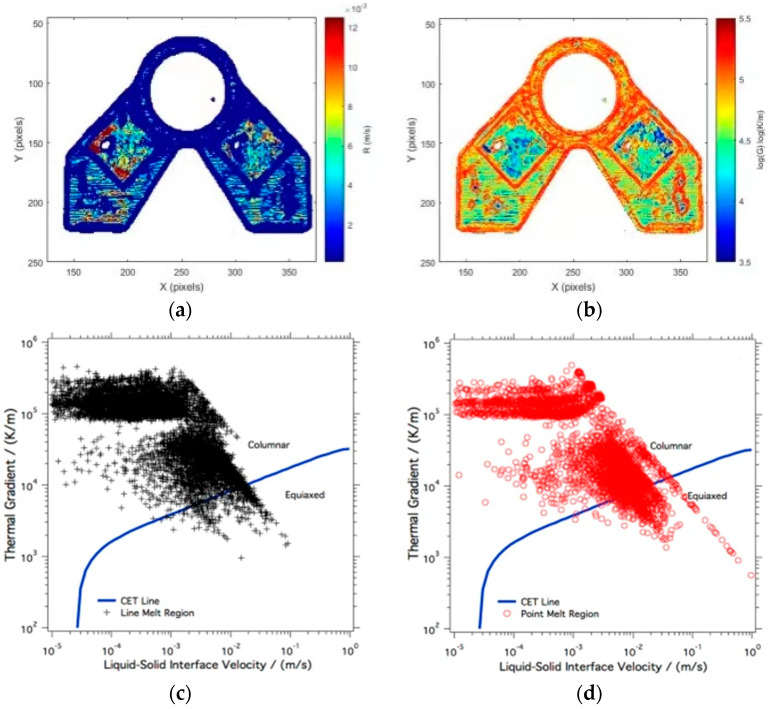
Influence of scan strategies on grain morphology in EBM [52]. (**a**) The layer thermal gradients; (**b**) Interface velocity; (**c**) Grain morphology distribution under line scan strategy; (**d**) Grain morphology distribution under point scanning strategy.

**Figure 14 materials-15-07598-f014:**
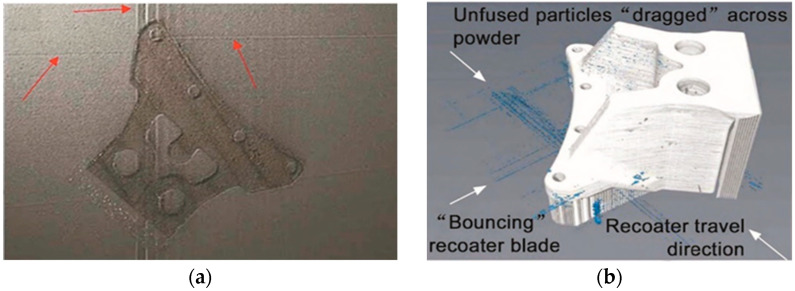
Melt layer inspection based on visual imaging. (**a**) Melt layer image; (**b**) 3D reconstruction model and powder bed anomaly [6].

**Figure 15 materials-15-07598-f015:**
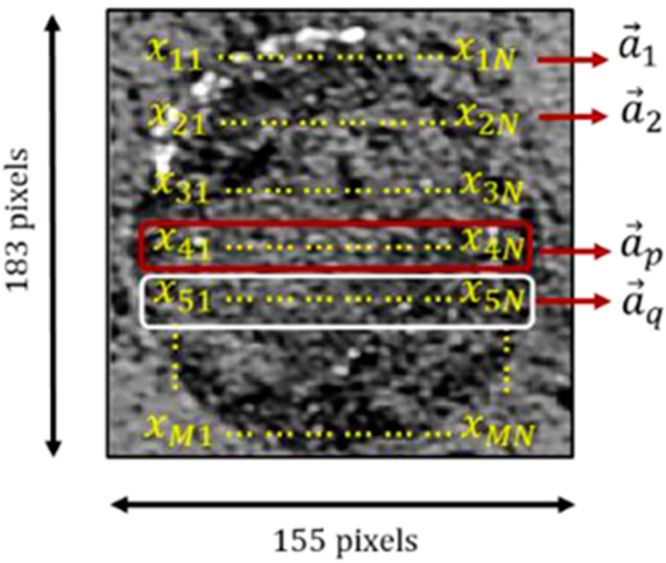
Diagram of linking features and process parameters using machine learning methods [62].

**Figure 16 materials-15-07598-f016:**
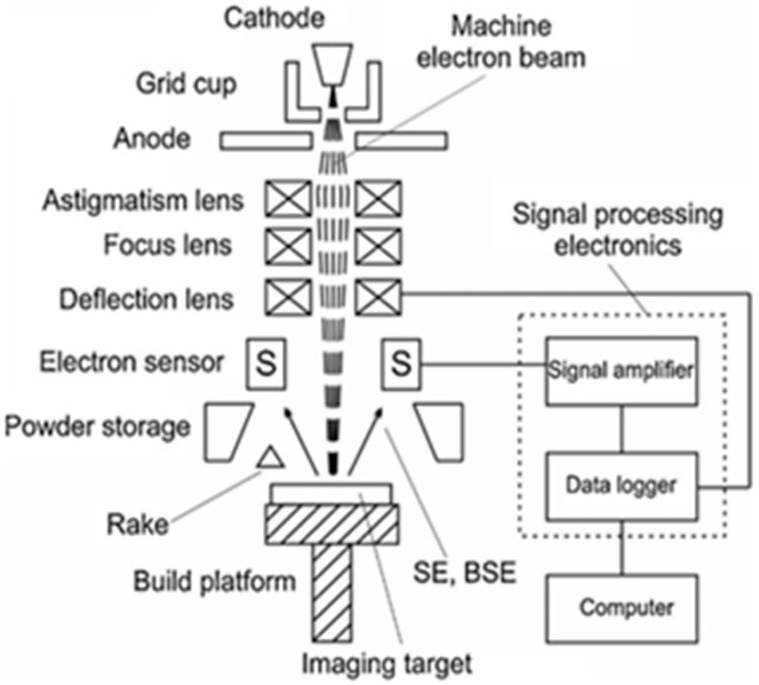
Electronic microscope schematic [63].

**Figure 17 materials-15-07598-f017:**
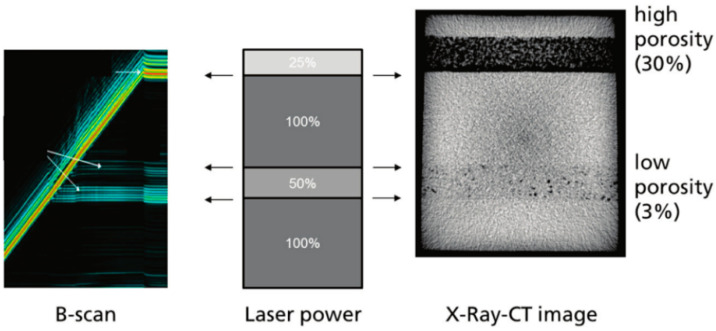
Ultrasound scan and corresponding X–ray CT scan [67].

**Table 1 materials-15-07598-t001:** Summarizes the powder recoating monitoring process.

Monitoring Process	Monitoring Method	Monitored Defects	Advantages	Disadvantages
Powder recoating monitoring	Digital camera	Coater problems, low or excessive powder feed	Low cost	Precision errors, requiring a trade–off between field of view and spatial resolution
	Piezoelectric accelerometer	Smoothness of coating, unevenness of previous deposited layer	-	-
	High–resolution CCD camera	Poor supports, coater damage, insufficient powder	Easy to operate	CCD cameras have a single use
	Edge projection profilometry	Powder overfeed, powder shortage, part thermal expansion	Low cost, no vacuum environment, fast acquisition time,	No automatic feedback, no intelligent measurement
	Powder scanner	Grooves, ultra–high edges, powder unevenness	High spatial resolution, automation	To ensure the synchronization of the recoater module movement and the CIS sample rate
The coating process is the key first step, but there is little information about the formation of defects.				

**Table 2 materials-15-07598-t002:** An overview of powder bed inspection.

Monitoring Process	Monitoring Method	Monitored Defects	Advantages	Disadvantages
Powder bed inspection	Visible light imaging	Damage to the coater	-	The exact location of the defect could not be obtained.
	Methods for thresholding grayscale images	Topological defects, powder bed defects	Parameter optimization (material identification)	The monitoring method based on optical imaging has higher requirements regarding the relative position of the sensor and the light source.
	Image–based two–dimensional acceleration	Influence of the angle of the overhang structure and the parameters of the support structure on the expanded melt layer	Sorting the stability of different components	
	Low–coherence interferometry	Powder bed flatness	-	
	Inline coherence imaging	Surface roughness, recoater blade damage, powder packing density	Correction of surface roughness based on ICI measurements, closed loop control, full feedback control	
	3D indexing	3D locations of powder bed anomalies	Accurate location of defects	-
	Digital image processing	Single–layer defects and defects between layers	Accurate, fast, and low cost	Image distortion
	EPMP	Inhomogeneities in powder beds, irregular surface of fusion area	Reliable, high precision, high efficiency	Failure to implement real–time closed–loop control or automatic defect identification and classification
	Camera layered acquisition combined with image processing	Powder deficiency, powder overload, powder bed contamination	Highly usable for industrial EBM	Cannot be used to monitor oxidation of surface powders
	Numerical simulation combined with GBNN	Thermal anomalies	Feedback control	-
	TS–CNN model	Warpage, short feed, part shifting	High precision, high efficiency, anti–geometric distortion	No real–time control
	MSCNN model	Debris, coater jumps, recoat streaks,	High anomaly classification accuracy	Lack of real time
	ML and DSLR camera	Recoater hopping, recoater streaking, debris, superelevation, part failure, incomplete	Less amount of calculation	Less accurate monitoring of repaint streaks
The powder bed is the basis of the melting process and effectively reflects the quality of current layer. Limited to monitoring the surface state of the current layer.				

**Table 3 materials-15-07598-t003:** Reveals building process monitoring.

Monitoring Process		Monitoring Method	Monitored Defects	Advantages	Disadvantages
Building process monitoring	Melt pool monitoring	Coaxial sensor	Warpage, spheroidization,	High local spatial resolution, high timeliness	With an on–axis setup, the laser can be affected by lens characteristics in the Lagrangian reference frame.
		Photodiode and CMOS camera	Spheroidization, convex hulls at corners, powder coating failures	Keeping the molten pool size constant by controlling the laser power, improving the forming accuracy of suspended surface structures	
		Low–coherence interferometry	Globular defects	High speed, real time	
		Location–based visual pore detection	Pore defects	Real time, high sampling rate	No feedback controls
		Thermal imaging and off–axis sensor	Pore defects, irregularities close to overhang structures	Widely used	Parameter deviation, no real–time control
		Numerical simulation combined with CMOS camera	Melting pool size	Low cost, feedback control	-
	Temperature monitoring	Near–infrared thermal imaging	Pore defects	Visibility	Insufficient temporal and spatial resolution, inaccurate temperature
		Two–wavelength pyrometer	Filling interval, filling strategy, thickness of powder layer	Sensitive to parameter changes	-
		Based on longitudinal temperature distribution	Internal void defects	Sensitive to heat dissipation conditions, defect size information can be obtained.	Evaporation, no real–time control, insufficient accuracy and sensitivity of defect detection
		Wide–field in situ infrared imaging	Pore defects	Error calibration, prediction of the microstructure of formed parts	-
Effectively reflecting the internal formation process. Contributing to real–time defect repair and organization control.					

**Table 4 materials-15-07598-t004:** Presents melt layer detection.

Monitoring Process		Monitoring Method	Monitored Defects	Advantages	Disadvantages
Melt layer inspection	Temperature detection	Infrared or near infrared	Pore defects, Unmelted material within the cambium, non–uniformity of temperature distribution	-	Lack of real–time control
	Surface topography detection	Visible light imaging	Inhomogeneous powder bed, internally not fused	Contour extraction, defect recognition	Analysis and processing of grayscale images, offline processing
		Low–coherence interferometry	Melt layer surface rough, suspended structure rough	Less analysis data	Long time, complex system, point–by–point scan
		3D mapping technology	Uniformity, thickness, layer defects		
		Electro–optical inspection	Pore defects, surface defects	Suitable for EBM, online monitoring, feedback control	Optical image research, roughness cannot be monitored
		Ultrasonic testing	Porosity	Correlated online and offline data	This method cannot be used for complex geometry.
		Acoustic emission spectroscopy	Spheroidization, slight spheroidization, slight overheating, overheating	-	-
		Spatially resolved acoustic spectroscopy	Porosity	-	PBF process signals are complex and must be integrated with ML.
		Acoustic emission spectroscopy and convolutional neural network	Porosity	Fast, efficient, positioning defects	-
Most intuitively reflecting the quality of melt layer.					

## Data Availability

The raw data supporting the conclusions of this article will be made available by the authors, without undue reservation.

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
