# Peer review of "Online Monitoring Technology of Metal Powder Bed Fusion Processes: A Review"

_materials, 2022, doi:10.3390/ma15217598_

Round 1

Reviewer 1 Report

Dear authors,

This manuscript addresses the on-line monitoring technologies for the metal additive manufacturing. The authors have reviewed well about  the recent progress of the on-line monitoring technoologies, and this review report is helpful to the researcher who have developed metal parts by metal additive manufacturing. Althogh the manuscript is well written, this need minor revision to be accepted.   1. This manuscript addresses the on-line monitoring for not only PBF, but also DED (ex. Figure 14 and

18). Then, I think that it is better to change the title. For example, replacing "Metal Powder Bed
Fusion" to "Metal Additive Manufacturing". In addition, it is better to mention DED in the introduction.

  Sincerely yours.

Reviewer 2 Report

The paper cannot be accepted for publication mainly due to its extremely low quality. This manuscript discusses 'everything and nothing" but about all related to the online monitoring technologies of metallic alloys by PBF technique.

The authors did not present the paper with a clear logic flow and focus; instead they presented too much information that is not needed, or with limited value, or should be presented in a concise way.

The important results are extremely limited.  Lot of results can be presented with a very short paragraph, instead of so many not-clearly-selected photos/figures.

The paper is lack of a good and deep discussion.

Because of the above I cannot recommend such review article for publication in acknowledged journal like Materials journal.

Reviewer 3 Report

The paper reviews on-line monitoring in metal powder bed fusion process. It is fully with the scope of the journal. The paper briefly reviews current research from four perspectives: powder recoating monitoring, powder bed inspection, building process monitoring and melt layer detection. This seems to be novel and vital. An adequate number of papers is addressed to cover the research from these four points of view. The papers are briefly discussed which in some places is insufficient and some details are needed. Figure 16 is not discussed in details the authors should describe it or exclude. Figure 14 shows laser metal deposition (LMD) online monitoring scheme. Though LMD systems for building process monitoring could be applied to PBF this figure is poorly described and misleading. Also the method described in ref 41 for LMD could be applied to PBF with precautions.

The numerical simulation coupling with on-line monitoring is not addressed in the paper, but coupled heat transfer models are known to be used for the process planning and calibration of the monitoring and feedback systems.

The perspectives of the discussed monitoring systems are not well shown. The shortcomings of current technology are not summarized. The defects that could be addressed by certain monitoring technic should be added to the table 1. The healing ability of the detected defects as one of the main purposes of the online monitoring should be summarized for the discussed monitoring methods.

The terms are not strictly used in the paper which puzzles the reader. For example the development trends are given for powder bed melting but not for powder bed fusion process. The former could be considered as a one of the perspectives which was addressed.  The paper contains numerous errata (example: lines 35, 229, etc.). The paper definitely needs English language improvement as some phases are not well posed (example: lines 243, 442, 451) or the text is hard to read due to for example word repetition.

The paper seems to have potential to contribute to the topic but major revision is needed.

Round 2

Reviewer 2 Report

The authors revised the paper named “Online Monitoring Technology of Metal Powder Bed Fusion Processes : A review”. I have reviewed the paper and see that the authors have been made many contributions to article. I think the paper is ready for publishing at the final version. My decision is about to accept it.

Reviewer 3 Report

There are still errata could be found in the text (line 437). Some parts of the text are taken from a thesis without any revision. It seems like there was no internal revision and proofreading. Some conclusions of the authors are not supported by the evidence from papers. For example the author does not understand the difference between monitoring of the defects and use temperature monitoring for defects identification. (“…the measurement and positioning of internal defects in parts has become a research focus. In order to achieve real-time defect repair and tissue control, this study started to involve the monitoring of grain structure”). This might be connected to the poor English. Some phrases could not be understood (line 88), some are not connected with the text (line 410 “In addition, there is evaporation in PBF processes especially in EBM process.”)

The numerical simulation usage for tuning the parameters of the monitoring and feedback systems is still not addressed in the sections and in conclusions.

The paper still lack analysis of the discussed monitoring systems. The ability to correct detected defects as one of the main goals of online monitoring should be generalized in the last section. The tables 2-4 should be discussed and conclusion should be driven of the discussion.

The tables are poorly readable; the font size reduction might work. And please highlight the certain changes of the text but not the whole sentences with changes.

Round 3

Reviewer 3 Report

There are still problems with construction of phrases and numerous errata. I leave this on the reputation of the authors.

Content highlighted in blue has nothing to do with numerical simulations in online monitoring. As the authors agree “After much discussion … this part to be very necessary” they should include this topic into the survey. For example they could use following paper where this topic is fully revealed.

Jahan, S. A., Al Hasan, M., & El-Mounayri, H. (2022). A framework for graph-base neural network using numerical simulation of metal powder bed fusion for correlating process parameters and defect generation. Manufacturing Letters33, 765-775.

Le, T. N., Lee, M. H., Lin, Z. H., Tran, H. C., & Lo, Y. L. (2021). Vision-based in-situ monitoring system for melt-pool detection in laser powder bed fusion process. Journal of Manufacturing Processes68, 1735-1745.

I also noticed that the authors ignored very interesting paper on digital  image processing in powder bed monitoring that could be included into the consideration.

 Boschetto, A., Bottini, L., Vatanparast, S., & Veniali, F. (2022). Part defects identification in selective laser melting via digital image processing of powder bed anomalies. Production Engineering, 1-14.

As to the rest of the paper I think that the job is done and it could be published in Materials journal after above mentioned minor revision.
